# Tenascin-C: A Key Regulator in Angiogenesis during Wound Healing

**DOI:** 10.3390/biom12111689

**Published:** 2022-11-15

**Authors:** Yucai Wang, Guangfu Wang, Hao Liu

**Affiliations:** 1Department of Orthopaedic Surgery, Tangdu Hospital, AirForce Medical University, Xi’an 710000, China; 2Vasculocardiology Department, The Fourth People’s Hospital of Jinan, Jinan 250000, China; 3Division of Vascular and Interventional Radiology, Department of General Surgery, Nanfang Hospital, Southern Medical University, Guangzhou 510000, China

**Keywords:** injury repair, tenascin-C, angiogenesis, extracellular matrix

## Abstract

(1) Background: Injury repair is a complex physiological process in which multiple cells and molecules are involved. Tenascin-C (TNC), an extracellular matrix (ECM) glycoprotein, is essential for angiogenesis during wound healing. This study aims to provide a comprehensive review of the dynamic changes and functions of TNC throughout tissue regeneration and to present an up-to-date synthesis of the body of knowledge pointing to multiple mechanisms of TNC at different restoration stages. (2) Methods: A review of the PubMed database was performed to include all studies describing the pathological processes of damage restoration and the role, structure, expression, and function of TNC in post-injury treatment; (3) Results: In this review, we first introduced the construction and expression signature of TNC. Then, the role of TNC during the process of damage restoration was introduced. We highlight the temporal heterogeneity of TNC levels at different restoration stages. Furthermore, we are surprised to find that post-injury angiogenesis is dynamically consistent with changes in TNC. Finally, we discuss the strategies for TNC in post-injury treatment. (4) Conclusions: The dynamic expression of TNC has a significant impact on angiogenesis and healing wounds and counters many negative aspects of poorly healing wounds, such as excessive inflammation, ischemia, scarring, and wound infection.

## 1. Introduction

One of the most complex and progressive molecular processes is wound healing, which involves hemostasis, inflammation, proliferation, and remodeling [1]. It is challenging to clearly identify the distinct stages of the wound healing process since each step of injury repair overlaps with another [2]. The various cell types and adaptive change mediators play a role in the healing process’s precise spatial and temporal regulation. The extracellular matrix (ECM) is crucial in the regulation of various growth factors and cellular functions in wound healing [3,4]. Tenascin-C (TNC), a hexamer ECM glycoprotein, was established in numerous studies to be essential for controlling cell inflammation, proliferation, migration, and angiogenesis [5,6,7].

First, TNC is only momentarily expressed during damage repair and tumor metastasis processes, not in normal tissues [8]. TNC is known to be synthesized by interstitial fibroblasts under inflammatory stimulation. TNC interacts with a number of other factors to create a pro-inflammatory positive feedback loop that generates more inflammatory response and recruits various inflammatory cells and growth factors at the site of damage. TNC itself is continuously up-regulated during the wound repair process. A large number of experiments have established that TNC exerts pro-angiogenic effects [9,10,11,12]. However, angiogenesis did not increase significantly during the inflammatory phase of tissue injury [13,14], although TNC levels gradually increased. One possible reason is that the heightened inflammatory response inhibits angiogenesis, concealing TNC’s pro-angiogenic role. As the restoration development proceeds, TNC expression levels continue to notably increase until the complete functionality of TNC has gradually unfolded. High levels of TNC cause a phenotypic change in macrophages from M1 to M2 polarization, which supports a progressive reduction in inflammation at the site of the injury. Without the inflammatory stimulus as the initiator, the TNC level can no longer increase and reaches a peak. At the same time, without the inhibition of inflammation, TNC demonstrates its strong pro-angiogenic effect. The last stage of the repair process moves on to the proliferation phase. Angiogenesis is decreased by TNC reduction. Subsequently, the wound starts to remodel.

Conclusively, we were shocked to find that TNC seems to have a substantial impact on the entire wound healing process and counteracts the negative effects of poorly healed wounds, including excessive inflammation, ischemia, scarring, and wound infection. Furthermore, we are surprised to find that post-injury angiogenesis is dynamically consistent with changes in TNC. Due to the wide range of clinical implications of TNC, this matricellular protein is a potential target for therapeutic intervention. These results provide a framework for preclinical investigation, with the ultimate goal of leading to the start of clinical trials.

## 2. Results

### 2.1. The Construction and Expression Signature of TNC

The ECM is mainly composed of various macromolecular substances that constitute a complex network of cellular physiological activities [15]. Recent studies have shown that ECM plays an increasing role in tissue damage and regeneration, especially in angiogenesis, by providing important cues for cellular responses [16,17]. Tenascins are multifunctional glycoproteins found in the ECM of chordate animals. Studies have validated that tenascin-C, -R, -X, and -W are the four isoforms of the tenascin family, and have similar functions in regulating cell adhesion and cellular responses to injury repair [18,19,20,21,22].

Since it was first discovered in the 1980s, TNC has been the most studied and most representative member of the tenascin family [5,23]. The unique structure of TNC often exists in the form of hexamers, with a molecular weight of approximately 220–400 kDa [24]. TNC consists of four distinct domains, from N-terminal to C-terminal: the coil-like tenascin assembly (TA) domain; 14.5 epidermal growth factor-like (EGF-L) domain; 17 fibronectin type III (FN III)-like domains; and a fibrinogen-like globule (FBG) domain (Figure 1A). This multi-modular structure enables tenascin to interact with a large number of highly diverse ligands, such as FNIII-like domains that bind different proteins, such as annexin II, syndecan-4, and integrins αvβ3 and α7β1 through alternative splicing [25,26,27]. In addition, FNIII-like domains bind to a variety of growth factors, including the vascular endothelial growth factor (VEGF), epidermal growth factor receptor (EGFR), and transforming growth factor-β (TGF-β). The interactions allow TNC to promote cell migration, cell adhesion, matrix aggregation, protease, and proinflammatory factor synthesis. Therefore, given the structural, expression, and functional characteristics of TNC, the alternative spliceosome of TNC plays different roles in various injuries (Figure 1B) [28,29,30,31,32,33].

In adults, the expression of TNC is strictly regulated by cell type and the tissue microenvironment. TNC is nearly undetectable in normal adults, with the exception of embryonic development and a few connective tissues. TNC mainly exists around motor cells, especially in the central nervous system and bone joint system [34]. TNC plays a role in regulating cell adhesion, motility, and tissue remodeling and is closely related to inflammatory signaling, through which angiogenesis can be achieved. Changes in TNC expression caused by external stimuli cause angiogenesis and are involved in the processes of trauma, tumor, and degeneration. Whereas in the pathological state, TNC exists around the injury, the tumor matrix, and the site of regeneration [35,36].

### 2.2. TNC in Healing Wound

Injury defined by damage to normal tissues is caused by multiple factors [37]. The integrity of the tissue is destroyed, along with the loss of a certain amount of normal tissue [38]. In the wound healing process, there are four relatively distinct phases that include hemostasis, inflammation, proliferation, and remodeling. The first step is the hemostasis phase of wound healing, which entails clotting of blood mediated by platelets [39]. Angiogenesis is a complex and dynamic process that plays an important role at each stage of wound healing by transforming vascular precursor cells into new blood vessels [40]. The generation of new capillary beds through angiogenesis is essential for normal healing [41]. Tissue vascular networks after injury usually involve angiogenesis, vasculogenesis, and arteriogenesis. Vasculogenesis is the process of growing new blood vessels from existing vasculature. Arteriogenesis is the process by which arteries remodel [42,43].

The ECM, including collagen, TNC, matrix metalloproteinases (MMPs), etc., are major players in tissue injury and regeneration, providing important regulatory clues for the formation of new capillaries [44,45]. Usually, angiogenesis increases greatly in the early stage of injury and will increase to 2–3 or even 10 times after injury [46,47]. In the later stages of repair, cell death and vascular degeneration return to normal levels to maintain homeostasis [48]. TNC is an important component of the ECM and is mainly expressed during tissue repair and tumor growth. The expression level of TNC is dynamically consistent with the changes in angiogenesis after injury in both time and space [48] (Figure 2). Studies suggested that the physiological repair of tissue injury can be mediated by regulating the dynamic expression of TNC.

Here, we sorted out the dynamic character of TNC during the course of the damage repair process and focused on the relationship between the dynamic changes of TNC and the injury repair process.

### 2.3. Initial Activation of TNC in the Inflammation Phase

Normal healed wounds provide an excellent model of strong capillary growth and controlled capillary regression. In adult tissues, TNC expression is very low [7]. The next step is the inflammation phase of wound healing, mediated by macrophages and neutrophils. At this stage, TNC is majorly secreted by fibroblast-specific protein-1 (FSP1)+fibroblasts in the entheseal inflammatory microenvironment [49]. The TNC gene promoter contains a TATA box [7,19], as is typical for highly regulated genes, as well as a variety of different transcription factors that help to regulate the various patterns of TNC expression. For instance, the product of the patterning gene-paired mesoderm homeobox protein 1 (Prrx1) directly regulates the TNC promoter [50]; it transactivates the mouse TNC gene by interacting with a conserved homeodomain-binding sequence in its promoter [51]. In contrast, homeobox even-skipped homolog protein 1 (Evx1) indirectly stimulates TNC expression by synergizing with transcription factors Jun and/or Fos [52]. At least two more homeobox transcription factors, Pou3F2 and Otx2, regulate TNC expression through direct binding to conserved sequences in its promoter [53,54]. Furthermore, proinflammatory cytokines up-regulate the expression of TNC, which may amplify inflammatory responses by creating a positive feedback loop [24,55]. In the inflammatory phase after myocardial infarction(MI), cardiomyocyte necrosis results in the release of molecular signals, damage-associated molecular patterns (DAMPs) [56,57,58], which trigger inflammation-driven fibrotic responses [59,60]. TNC functions as a DAMP and is one of the ligands for the toll-like receptor 4 (TLR4) transmembrane protein [58,61,62]. TNC induces the synthesis of proinflammatory cytokines through TLR4 [61,62,63,64,65,66,67] and up-regulates the expression of MMPs in various cell types, including macrophages [65,68]. In vitro, TNC/TLR4 enhances the M1 macrophage polarization of bone-marrow-derived macrophages but inhibits the up-regulation of the M2 macrophage marker by suppressing interferon regulatory factor 4 (IRF4) [64]. TNC also accelerates the migration and up-regulates the synthesis of proinflammatory cytokines/chemokines via integrin αvβ3 in peritoneal macrophages [69].

Thus, the TNC-mediated positive feedback loop can promote TNC expression significantly in the area of injury. Moreover, in the subsequent phases, peak TNC is considered the basis of physiological damage repair and pathological remodeling.

### 2.4. Peak Expression of TNC in the Proliferation Phase

The proliferation phase of wound healing involves the recruitment of a large number of macrophages, lymphocytes, fibroblasts, and keratinocytes to repair the damage [70]. Researchers generally accept that the critical elements during this stage are multiple dynamic angiogenic responses and large amounts of pro-angiogenes. The peak expression of TNC is involved in directly and indirectly accelerating wound repair and tumor metastasis through a variety of mechanisms, including pro-angiogenesis, M2-type polarization of macrophages, changing growth factors, peptides, and fiery miRNA clusters [26,71,72,73] (Table 1).

#### 2.4.1. TNC Promotes the M2 Polarization of Macrophages

TNC stimulates M1 phenotype macrophages and is pro-inflammatory during the inflammatory phase of trauma [74,75]. However, during the proliferation phase, TNC exhibits different effects. At this stage, macrophages are mainly transformed from pro-inflammatory to anti-inflammatory, that is, polarization from M1 to M2. M2 macrophages produce and release large amounts of anti-inflammatory and immunosuppressive factors, such as Interleukin 10 (IL-10), which suppresses inflammatory responses while promoting cell growth, migration, and angiogenesis. Wang’s results showed that in an atherosclerosis model, the expression of TNC increased due to inflammatory stimulation [71]. Increased expression of TNC can increase cell movement, migration, and angiogenesis. TNC mainly binds to its receptor annexin II to activate the Akt/NF-κB and extracellular-signal-regulated kinase (ERK) pathways, thereby increasing VEGF expression. Inhibition of macrophage annexin II expression or TNC content inhibits Akt/NF-κB and ERK pathways and reduces production of vascular endothelial growth factors (VEGFs), thereby reducing cell migration and angiogenesis. In a three-month-old mouse model of MI, the left ventricular ejection fraction was significantly increased after TNC overexpression, which protected the heart from adverse remodeling, and macrophages underwent polarization from M1 to M2 [64]. In chronic MI, increased expression of TNC promotes the polarization of M2 macrophages, a large number of new blood vessels are formed, and the symptoms of patients are relieved. In a TNC-knockout (TNC KO) mouse model of corneal injury, neovascularization from the marginal region to the central cornea was inhibited by blocking p38/MAPK and Smad3 signals [30]. As a receptor for TNC, angiogenesis is inhibited by integrin α9 inhibition, which is associated with M1 macrophage invasion. M1 macrophages inhibited VEGF and transforming growth factor β (TGF-β) expression. The results showed that a rising level of TNC or integrin α9 could promote angiogenesis after corneal injury. TNC primarily inhibits M1 macrophage invasion and promotes macrophage transformation from the M1 to M2 phenotype [64].

These results suggest that TNC plays different roles at various stages of damage repair. This is an interesting phenomenon that calls for more research to fully understand the molecular mechanism of the transfer function caused by dynamic TNC changes.

#### 2.4.2. Associations between TNC and VEGF and TGF-β and Angiopoietin

The VEGF family comprises secreted proteins critical for regulating vascular permeability, vasodilation, and angiogenesis, and is proven to stabilize the growth of new blood vessels during wound repair [76]. After tissue injury, VEGF produced by local cells can diffuse into the bloodstream, resulting in increased VEGF concentrations in systemic circulation. During tissue remodeling, high levels of VEGF stabilize the growth of new blood vessels and provide a prolonged angiogenesis signal cycle [77]. In a breast cancer mouse model, fibroblast-derived TNC increased the expression of VEGF-A, resulting in increased angiogenesis and progression of breast cancer [78]. As a checkpoint of VEGF action, the inhibitor of VEGF, TNC, significantly reduced the probability of lung metastasis of breast cancer and increased mouse survival time [79]. Clinical studies have shown that in patients with proliferative diabetic retinopathy (PDR), new blood vessels show negative change after injecting anti-VEGF drugs [80]. Therefore, TNC is very important in regulating the physiological changes mediated by VEGF during trauma and, additionally, in promoting cell proliferation and angiogenesis.

TGF-β is another important mediator of angiogenesis and tissue remodeling. TGF-β promotes mesenchymal stem cell proliferation and differentiation, as well as vasoconstriction and angiogenesis [81]. A recent study has shown that TGF-β interacts with TNC to regulate angiogenesis [82]. Hematogenous tumor metastasis is an important cause of death in patients with advanced cancer. A study on breast tumors showed that TNC could increase lung metastasis in breast cancer. However, the chance of breast cancer lung metastasis was significantly reduced in TNC KO mice [83]. TNC promotes breast cancer lung metastasis primarily by activating TGF-signaling, which improves cell plasticity, angiogenesis, cell survival, and migration. Choroidal neovascularization (CNV) and macular degeneration were found in breast cancer patients with metastases [84,85]. The pathogenesis may be related to the pro-angiogenic function of TNC. The results of animal experiments corroborate this conjecture. In a mouse model of CNV, a marked increase in TNC expression was observed, while TNC binding to integrin αV receptors regulated the proliferation, adhesion, and tube-forming abilities of human microvascular endothelial cells (HMVECs), thereby promoting angiogenesis [86]. TGF-β2 increases the expression of TNC and enhances the proliferation and tube-forming ability of HMVECs. The proliferative tube-forming abilities of HMVECs were greatly reduced after treatment with TNC KO mice or with TNC siRNA. Therefore, TNC alters TGF-β signaling and causes neovascularization, leading to tumor blood metastasis and aggravation of macular degeneration.

Angiopoietin (Ang) is the first identified cytokine derived from human tumor tissue and has the effect of promoting angiogenesis [87,88]. Its main function is to regulate the development and stability of the vascular system. Ang can mediate the migration, adhesion, and survival of endothelial cells and plays a role in mediating the connection between endothelial cells and perivascular cells. Therefore, Ang plays a key role in angiogenesis during wound healing. Myocardial infarction (MI) is primarily caused by myocardial cell ischemia [89]. Angiogenic therapy is important for the survival and prognosis of MI. TNC and Ang-2 levels were significantly elevated in a rat model of MI [90]. TNC and Ang-2 levels were significantly increased in the oxygen and glucose deprivation (OGD) model [90]. TNC levels decreased after histone methylation modification, and Ang-2 reversed this decrease. In hypoxic H9c2 cells of embryonic rat cardiomyocytes, the levels of pro-angiogenic molecules such as BNP, MMP-2, and TNC were significantly increased after treatment with Ang. Ang levels also increased after treatment with TNC. Therefore, after MI, Ang-2 and TNC promote each other’s expression and increase the formation of new blood vessels.

#### 2.4.3. TNC and miRNA Cluster

Gene regulation plays a key role in angiogenesis during wound healing. With the ability to increase or inhibit hub gene expression, miRNAs have become a focus in wound healing [91,92]. A few miRNAs (mir21, mir-23a, mir-26a, and mir-27b) were shown to specifically regulate angiogenesis during wound repair [93,94,95,96]. These miRNAs are involved in increasing cell movement, regulating migration, reducing scar formation, and regulating the pro-angiogenic proteins VEGF, Ang, and TGF-β, as well as the anti-angiogenic proteins PEDF and CXCR3. Recent studies have made it increasingly clear that miRNAs play a powerful role in regulating angiogenesis during wound healing.

In glioma tissues and cells, the expression of mir-107 and mir21 is down-regulated, whereas overexpression of mir-107 can inhibit the migration, invasion, and tubulation of glioma cells by directly targeting Notch2 expression [97,98]. Notch2 is known to activate TNC and cyclooxygenase-2 (COX-2). The results have shown that miRNA clusters could promote the expression of TNC and COX-2 by activating Notch2, thereby increasing the angiogenesis and invasion of gliomas. Another study on colon cancer metastasis showed that the increase in mir-198 level could promote an increase in TNC expression, enhance tumor angiogenesis and migration, and contribute to the progression of liver metastasis of colon cancer [72]. Therefore, TNC can influence miRNA cluster changes to regulate tumor angiogenesis.

### 2.5. TNC in the Remodeling Phase

With the resolution of inflammation, TNC expression gradually decreases in the late stages of injury. The dynamic reduction in TNC, which decreases the angiogenic-related response, is an important marker of entering the remodeling phase. Over time, most neovascularization degenerates in the late stages of injury until the final vascular density returns to the state of normal uninjured skin.

Moreover, molecules related to TNC play essential roles in tissue remodeling during the late stage of injury repair, which are further discussed below.

#### 2.5.1. TNC and PEDF

Pigment epithelium derived factor (PEDF) is a glycoprotein with a molecular weight of 50 kDa [99]. It is the most effective neoangiogenesis inhibitor [100]. The anti-angiogenic efficacy of PEDF is selective because it targets neovascularization and is almost ineffective for existing vessels [101]. Thus, PEDF is of great importance for accelerating the wound healing process.

Loss of PEDF can promote angiogenesis in retinal endothelial cells [102]. Flow cytometry, MTT assay, and transwell migration assays have shown that PEDF deletion is mainly associated with increased cell proliferation, decreased apoptosis, and increased angiogenesis. Molecular studies show that the mechanism of PEDF deficiency promoting angiogenesis is primarily related to increased levels of TNC, cohesin, thrombospondin-1, collagen IV, and decreased osteopontin content [103]. Therefore, in the process of vascular degeneration in the later stage of trauma, the increase in anti-inflammatory levels reduces TNC expression and promotes an increase in PEDF levels, thus, inhibiting the process of angiogenesis.

#### 2.5.2. TNC and CXCR3

The chemokine receptor CXCR3 is a member of the G-protein coupled receptor family and is mainly expressed in the parenchymal cells of damaged organs and inflammatory cells [104]. Since CXCR3 can bind to its ligands CXCL9, CXCL10, and CXCL11 to inhibit angiogenesis, CXCR3 may play an important role in wound healing and tumor growth [105].

A recent study has shown that transplantation of mesenchymal stem cells (MSCs) can promote wound healing by regulating inflammatory responses, inducing angiogenesis, and enhancing epithelial growth and extracellular matrix formation [106]. The pro-angiogenic ability of MSCs can be accelerated by the use of type I collagen and TNC hydrogels. MSCs mainly inhibit the expression of CXCR3 in fibroblasts, thereby reducing the formation of hypertrophic scar tissue and regulating angiogenesis. The removal of TNC-containing hydrogels significantly inhibited the effects of MCCs. During the post-injury scarring phase, TNC may modulate the post-injury repair ability by increasing the level of CXCR3 in fibroblasts and reducing capillary formation.

## 3. Discussion

Long-term evolutionary forces have created what we are today, a highly adaptive multicellular organism with self-protection from daily chemical, physical, and ultraviolet radiation. Nevertheless, trauma, surgery, and various diseases can break this seemingly strong fortress and affect the health of tens of millions people worldwide. Wound healing is a universal aspect of medical care, with approximately 300 million chronic and 100 million traumatic wound patients worldwide [2]. Wound care has an immense financial burden on health-care systems worldwide, accounting for over $25 billion every year in the United States alone. In addition, the incidence of chronic wounds has rapidly increased due to the rising prevalence of type 2 diabetes, peripheral vascular disease, and metabolic syndrome. Although treatments for acute and small area traumatic wounds are effective, problems arise in the long-term care of patients with large area burns, infected wounds, and chronic wounds.

Data collection on cellular and molecular biology, immunology, and physiology allows us to learn more about the biological functions of TNC through changes in angiogenesis following trauma. After reviewing the expression characteristics of TNC and the changes in vessels after trauma, it was surprising that dynamic changes of TNC after trauma can be involved in the whole process of injury repair by regulating the inflammatory response, pro-angiogenesis, and remodeling. Due to the extensive role of TNC in inflammation regulation and damage repair processes, it is obvious that interventional therapies targeting TNC bear potential application prospects. In fact, blocking TNC has shown significant efficacy for healing disturbances or pathological damage caused by excessive local inflammation. Furthermore, it is difficult to heal injuries with severe mechanical damage or large-area burns. Unfortunately, the risk of infection is also greatly increased without the skin barrier. A reasonable TNC supplement can regulate inflammation, control infection, and promote damage repair.

Accordingly, it would be of interest to further investigate the molecular mechanism underlying the regulations and functions of TNC with the aim of developing a clinical trial to evaluate the clinical potential of TNC as a therapeutic target for wound repair.

## Figures and Tables

**Figure 1 biomolecules-12-01689-f001:**
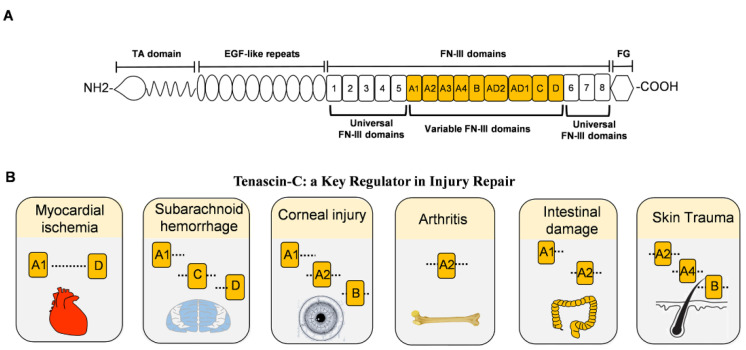
The structure of TNC and damage-related variable FNIII domains. (**A**). Full-length domain of human TNC protein. From the N-terminal to the C-terminal includes the tenascin assembly domain (TA), epidermal growth factor-like (EGF-L) repeat, fibronectin type III (FN III) domain, and fibrinogen globulin (FG) domain. The FNIII domains include eight fixed FNIII domains and nine variable FNIII domains. (**B**). The role of the variable FNIII domains in different injuries. All variable FNIII domains play a role in myocardial infarction. A1, C, and D domains aggravate the symptoms of subarachnoid hemorrhage. Corneal injury exhibits A1, A2, and B domains abnormally, while arthritis expresses the A2 domain leading to synovial injury. In intestinal damage, A1 and A2 domains aggravate intestinal bleeding. For skin trauma, domains A2, A4, and B are associated with angiogenesis.

**Figure 2 biomolecules-12-01689-f002:**
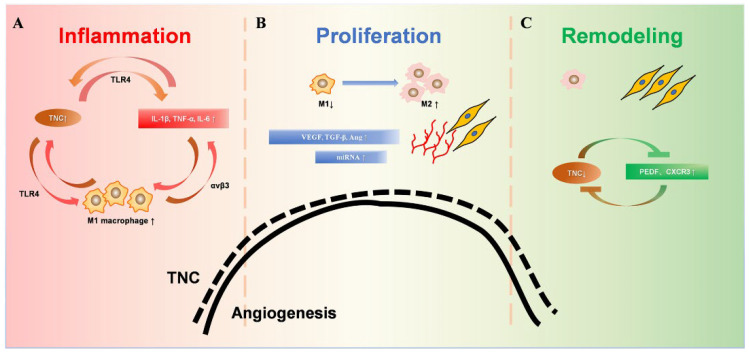
Dynamic changes of TNC during wound healing. (**A**). During the inflammatory phase, TNC induces the synthesis of proinflammatory cytokines through TLR4. TNC enhanced the M1 macrophage polarization of bone-marrow-derived macrophages. Furthermore, proinflammatory cytokines up-regulated the expression of TNC significantly, which amplifies inflammatory responses by creating a positive feedback loop. (**B**). The proliferative phase is mainly granulation tissue, including a large number of new capillaries, fibroblasts, and M1 and M2 macrophages. The substantial increase in TNC promotes angiogenesis, accompanied by promotion of macrophage polarization into the M2 phenotype. Increased M2 phenotype macrophages promote collagen formation and extracellular matrix proliferation. (**C**). During the remodeling stage, the decreased TNC expression leads to decreased M2 phenotype macrophages. Increased fibroblasts and collagen fiber formation are involved in tissue remodeling. Capillaries atrophy and decrease, and TNC decreases to the pre-injury state.

**Table 1 biomolecules-12-01689-t001:** Major factors regulating injury repair.

Mediators	Receptors/ Ligands	Functions
Pro-Angiogenesis Phage		
VEGF	VEGFR-1, VEGFR-2	Cell proliferation, migration, vascular permeability
TGF-β	TβR	Cell proliferation, differentiation; HIF expression
SDF-1	CXCR4	Cell survival, proliferation, chemotaxis
Ang	TIE2	Cell adhesion, migration, and homing
miRNA		Cell movement, migration
Anti-Angiogenesis Phage		
PEDF	PEDFR	Cell adhesion, migration, differentiation
CXCR3	CXCL9/10/11	Cell growth, chemotaxis

## Data Availability

Not applicable.

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
