# Peer review of "Tenascin-C: A Key Regulator in Angiogenesis during Wound Healing"

_biomolecules, 2022, doi:10.3390/biom12111689_

Round 1
Reviewer 1 Report
in this review, Yucai Wang et al. describe the role of tenascin-c in angiogenesis during wound healing. the manuscript is well organized, easily readable and can be understood also by non-expert reader. In my opinion, the review can be acceptable in the present form
Author Response
Response to Reviewer 1 Comments
Response 1: Thank you for your hard work and recognition of our research. We look forward to having more academic exchanges with you in the future.
Reviewer 2 Report
Overall it is a good review, focusing the role of TNC in wound healing and angiogenesis.
The exact reason for the TNC expression pattern changes was not explained at the molecular levels. That is, how TNC expression is regulated?
Minor: line 233-234: sentence is not complete. .....,Breast cancer also .......
Author Response
Response to Reviewer 2 Comments
Point 1: The exact reason for the TNC expression pattern changes was not explained at the molecular levels. That is, how TNC expression is regulated?
Response 1: Thanks to your professional advice, the regulatory mechanism of TNC was added to the manuscript. line 153-164:
The TNC gene promoter contains a TATA box [7, 19] , as is typical for highly regulated genes, as well as a variety of different transcription factors that help to regulate the various pat-terns of TNC expression. For instance, the product of the patterning gene paired meso-derm homeobox protein 1 (Prrx1) directly regulates the TNC promoter [50] it transactivates the mouse TNC gene by interacting with a conserved homeodomain-binding sequence in its promoter [51]. In contrast, homeobox even-skipped homolog protein 1 (Evx1) indirectly stimulates TNC expression by synergizing with transcription factors Jun and/or Fos [52]. At least two more homeobox transcription factors, Pou3F2 and Otx2, regulate TNC expression through direct binding to conserved sequences in its promoter [53, 54]. At least two more homeobox transcription factors, Pou3F2 and Otx2, regulate TNC expression through direct binding to conserved sequences in its promoter [55].
Point 2: Minor: line 233-234: sentence is not complete. .....,Breast cancer also .......
Response 2: Thanks for the reminder, we have made changes and added more references to address this issue. This sentence was revised to read: "Choroidal neovascularization (CNV) and macular degeneration have been found in breast cancer patients with metastases. The pathogenesis may be related to the pro-angiogenic function of TNC. The results of animal experiments corroborate this conjecture."